# Historical Wines of Portugal: The Classification, Consumer Associations and Marketing Implications

**DOI:** 10.3390/foods10050979

**Published:** 2021-04-29

**Authors:** Ana Isabel de Almeida Costa, Carla Marano-Marcolini, Manuel Malfeito-Ferreira, Virgílio Loureiro

**Affiliations:** 1Católica Lisbon School of Business & Economics, Universidade Católica Portuguesa, Palma de Cima, 1649-023 Lisboa, Portugal; anacosta@ucp.pt; 2Departamento de Organización de Empresas, Marketing y Sociología, Campus Las Lagunillas s/n, Universidad de Jaén, 23071 Jaén, Spain; 3Linking Landscape Environment Agriculture and Food (LEAF) Research Center, Instituto Superior de Agronomia, Universidade de Lisboa, Tapada da Ajuda, 1349-017 Lisboa, Portugal; mmalfeito@isa.ulisboa.pt (M.M.-F.); virgilioloureiro@gmail.com (V.L.)

**Keywords:** historical wines, brand authenticity, wine knowledge, consumer behavior, marketing strategy

## Abstract

Geographical origin, use of traditional varieties and ancestral viticulture/oenology practices characterize wines classified as Historical Wines of Portugal (HWP). This study identifies the authenticity attributes consumers associate with this classification and assesses the relative strength of associations. A review of brand authenticity research and interviews with Portuguese wine producers (*n* = 3) and consumers (*n* = 12) were conducted to identify HWP classification attributes. Strength of attribute association was subsequently evaluated in an online questionnaire with a convenience sample of Portuguese wine consumers (*n* = 641), which included a measure of general wine knowledge and questions about the adequacy of different contexts for HWP purchase and consumption. Wine knowledge markedly affected the nature and strength of consumer associations. Compared to *Aspirational Explorers*, wine *connoisseurs* emerged as *Heritage Gatekeepers*, associating origin, cultural heritage, quality, production and at-home consumption more strongly with HWP, and tradition, wine age and out-of-home consumption less strongly. Market recognition of HWP as a novel and distinctive table wine classification, with well-defined and unique attributes, is thus likely to depend on consumers’ general wine knowledge. Related promotional activities targeting wine novices should first focus on educating them on HWP classification, whereas those directed at savvier consumers should emphasize wine authenticity cues instead.

## 1. Introduction

### 1.1. The Portuguese Wine Industry

Portugal has an important winemaking sector, currently holding the seventh largest vineyard area in the world and ranking first globally in terms of dedicated share of total planted agricultural area (14% over 200.000 ha). Wine production in this country reaches ca. 7 million hL/year, making it the 11th largest wine producer worldwide. The sector therefore has a major economic impact on the country. Portugal is the 10th largest wine exporter in the world, with wine sales contributing significantly to the value of national exports and sustaining many related business activities, particularly in the food retail, food service and tourism industries [1]. Meanwhile, and after a few decades of steady decline, table wine consumption in Portugal has been slowly recovering in recent years due to improved economic prospects and record tourism growth [2]. Nevertheless, traditional, small local producers still struggle to be recognized and appreciated amongst the rising number of Portuguese wine brands. On their success often hangs the survival of family businesses and rural communities, as well as that of Portugal’s immensely rich and nuanced (though often grossly undervalued) genuine wine culture [3].

### 1.2. The Historical Wines of Portugal Classification

The Association of the Historical Wines of Portugal (AVHP) was born in 2008, with the aim of protecting and preserving Portuguese wine culture and heritage. This was the first body to propose the creation of *Vinhos Históricos de Portugal*, or the Historical Wines of Portugal (HWP), classification. According to AVHP, a combination of geographical origin, use of traditional grape varieties and application of ancestral viticulture/oenology practices characterize HWP and their production. Examples of these wines (Figure 1) are *Vinho Verde Enforcado* (Minho), *Palhete Medieval de Ourém* (Encostas D’Aire-Lisboa), *Vinho do Chão de Areia de Colares* (Sintra-Lisboa), *Vinho da Talha/Vinho Petroleiro* (Alentejo) and *Vinho do Pico* (Pico-Azores) (See Appendix A for descriptions) [3,4,5,6].

Historical wines have evolved as a product and a symbol of civilization, brought by the Phoenicians and Romans to the Iberian Peninsula, to become living recollections of past vines and winemaking techniques in territories of Portugal and Spain [3,7], as well as true statements of identity for local populations today [8]. Benefiting from the inputs of other cultures passing by or settling in the Peninsula over the ages, they were necessarily shaped further by local natural resources (indigenous grape varieties) and constraints (landscape and climate). Hence, along with wines from the “heroic viticulture”, still practiced in Italy, for instance, particularly in the region of Sicily [9], historical wines represent one of the most accomplished embodiments of the concept of *terroir* outside France.

As with other food heritage products [10], ancestral ways of making historical wines have been preserved until today due to the relatively slow economic development observed in some areas of wine-producing countries in the Mediterranean [7]. AVHP efforts to implement an HWP classification represent a first attempt to create a wine brand that combines the preservation of rural memories of local communities with the cultural and potential market appeal of food products of ancestral tradition [11], in an effort to target the modern consumer eager to reconnect with food, place and culture in an era of globalized agri-food systems [12].

### 1.3. Branding Authenticity in the Wine Industry

Branding authenticity is becoming increasingly relevant for the marketing of a wide range of food products and related services that hold potential for the creation of meaning to consumers [12,13]. A prime example of this is wine and the broad range of production, consumption and leisure activities that surround its production [14]. At the same time, classification schemes qualifying foods and beverages as a genuine version of a product in relation to a *terroir*, such as the European Union’s PDO/PGI (Protected Designation of Origin/Protected Geographical Indication), raise the issue of what producers and consumers consider to be a genuine or authentic product in the first place [15].

Beverland [16,17] examined the meanings of brand authenticity in the premium wine market. He identified six dimensions of authenticity—heritage and pedigree, stylistic consistency, quality commitments, relationship to place, method of production and downplaying commercial motives—shared both by wine producers and consumers that conveyed an image of integrity and sincerity. These dimensions have considerable overlap with the defining features of wines classified as HWP [3].

Earlier research indicates that the presence of a product over time, and it gradually becoming part of the collective memory as an exemplar of the heritage and tradition of a place, are two important drivers of consumers’ perceptions of food authenticity [15]. History and tradition confer heritage to wines by referring to symbolic or non-functional product associations, such as being the first to have pioneered a certain production method or grape variety [17]. Many European wine producers excel at this because they own centuries-old wineries that are often handed down for multiple generations. Indeed, when a wine label is able to live up to its heritage of high-quality production, it gains stature or “pedigree” [18]. Heritage character and the use of ancestral viticulture and/or oenology knowledge and practices are also some of the defining features of historical and “heroic” wines, which may thus help associated brands build status and reputation [3,11].

Relationship to place is equally highly relevant for claims of wine authenticity [12,19]. In French viticulture, there is the related concept of *terroir* [20], which addresses the interplay between the quality, taste and style of wines and their geographical origin, including both natural and human environment factors. The broader concept of *terroir* implies the grounding of a product in a geographical place and the existence of specific representations for it in consumers’ minds, which relate to local history, culture and know-how [21]. That is, the land where foods grow or the region where wines are produced are assumed to impart them with unique and valuable features. Such relationship to place appears to be assessed by consumers essentially through the geographical origin of a product (or merely of its ingredients or methods of production), often with some help from formal certification schemes and labels, as well as from authoritative authentications of genuineness emitted by experts or media gatekeepers [15,22,23]. Charters and colleagues [24] suggest, furthermore, that *terroir* is a valuable marketing communication resource “based on unique physical origins and shared cultural personification that shape a product’s benefits into a meaningful value proposition not possible for products lacking this specific origin”. *Terroir* and local production methods are believed to combine to create the “body and soul” of ancient wines, conferring the distinctive character that sets them apart from other offers in the marketplace and granting them authenticity [9].

The organic and symbiotic character of the relationship between the wines classified as HWP and their regions of origin, with its unique physical and human landscape and its ancient history and traditions, is one of the key factors conferring quality and distinctiveness to these products [3]. Meanwhile, large-scale manufacture and mass-marketing of food products, the making of which entails the use of raw materials from different and often distant provenances, are increasingly perceived by consumers to “dissolve” the connection to place and, hence, to downgrade product quality and authenticity [12]. In line with this, there seems to be a trend for discerning consumers to value wines that are scarce, made from a single, local grape variety and sold through exclusive distribution channels, i.e., not mass-marketed [18,25]. Albeit largely unintendedly, wines classified as HWP share some of these important features due to a historical lack of access to outside resources in the areas they are produced, the small scale of their production and the largely artisanal, family character of the businesses involved in their distribution and retail sale [3].

Last but not least, claims of authenticity should be backed by wineries’ efforts to remain true to their past, namely by retaining the stylistic consistency and quality of their wines across decades [17]. Alongside heritage and sincerity, distinctiveness and long-term commitment to quality seem to be key to grant authenticity in the eyes of consumers [26]. Continuity of style in the face of evolving market preferences signals a desire to remain true to the brand and its values, as does the amount of time and money invested in guaranteeing that standards of production and end-product quality remain high over the years. The premium prices charged by brands claiming authenticity both reflect and reinforce these efforts [18]. Widely spread symbols of distinctiveness and quality commitment in the premium wine industry are the choice of high-quality wood, typically American or French oak for vats, or the exclusive use of the winery’s own grapes and the care put into their selection [16].

In view of longstanding economic constraints, wines classified as HWP have found other ways to signal stylistic consistency and quality commitment that possibly reinforce their claim of authenticity, such as the ability to resist modern market trends and remain true to their essence throughout the years, and a commitment to high artisanship and exclusive use of manual labor [3]. Earlier research uncovered that the use of unprocessed, local ingredients and manufacturing processes evocative of artisanal or craft production (e.g., handmade, homemade), are indeed likely to reinforce consumers’ perceptions of the quality and authenticity of food products [15,22].

### 1.4. Effects of Consumer Wine Knowledge

Searching, choosing and appreciating wine is a complex food consumption act that requires skill and effort. The levels of product knowledge and purchase involvement of different consumer groups should thus be taken well into account during the development of wine marketing strategies [27]. Wine brands should assess these consumer dimensions accurately and translate results into valid and actionable targeting and positioning efforts for each segment identified accordingly [28].

Consumer wine knowledge is a construct involving two main dimensions: familiarity (number of accumulated wine consumption experiences) and expertise (ability to perform tasks related to wine purchase and consumption, e.g., searching for product information or selecting a wine for someone else) [29]. Wine expertise can be assessed through both subjective and objective measures. Subjective wine expertise is, to a large extent, associated with consumers’ self-confidence about making wine consumption decisions, whereas objective wine expertise is more linked to their ability to process and apply product attribute information in decision-making [30,31]. Consumer levels of subjective and objective product knowledge have been shown, for instance, to affect how sources of wine marketing information are used [27,32].

Depending on their levels of product familiarity, subjective and objective expertise, consumers can be divided into novice (low knowledge) and expert (high knowledge) segments [30]. Important differences in the source, amount, content, acquisition, organization and use of product knowledge have been uncovered between these segments [27,31,32]. Experts were shown to attach more value to the geographical origin of wine, the *terroir* and other product-related characteristics such as vintage, than novices [29,33]. They also buy wine more frequently and are willing to pay more for it [34]. Contrary to novices, experts also tend to prefer wines that are not too sweet in taste or soft on the palate [35]. Importantly, previous research suggests that product familiarity and expertise might affect consumers’ perceptions of food and wine authenticity by shaping their beliefs and expectations about what an authentic product should be [36,37].

Degree of product involvement, that is, how personally relevant the consumption of a product is for an individual, has been shown to interact with product knowledge to determine the cues consumers use to evaluate and select wines [38]. Hence, both constructs have been combined to segment wine consumers in markets of different countries [27,39]. Early studies of the Australian wine market identified distinct groups of consumers according to their degree of involvement with, and knowledge about, wine [40,41]:The *wine connoisseurs*, who are enthusiastic, frequent consumers of fine wine, with high product expertise;The *aspirational drinkers*, who are equally enthusiastic but less knowledgeable and seasoned wine drinkers than *Connoisseurs*, and who are, hence, highly risk-averse consumers keenly interested in listening to, and learning from, opinion leaders about wine;The *beverage drinkers*, representing the no-frills wine consumer with high product familiarity, but not necessarily high expertise, who is mainly interested in the hedonic aspects of wine consumption;The *new drinkers*, who are consumers with little knowledge of, and low involvement with, wine drinking, and whose consumption is scarce and very much driven by context (place, company and occasion), namely by what family members and peers consume or advise.

Segmentation of consumer wine markets based on product knowledge and involvement has consistently yielded important differences in terms of how expert versus novice consumers carry out information searches or take the recommendations of authoritative sources, opinion leaders and peers into account when selecting and buying wines [41,42]. Thus, it is likely that the importance and credibility attributed to different cues related to wine authenticity will vary considerably across such segments.

### 1.5. Study Aims

Wine purchases demand a high cognitive effort from consumers and represent a large spending risk, unlike those of most other foods [30]. Wines are widely heterogeneous and complex beverages [43], and their quality cannot be easily evaluated prior to first purchase or tasting [44]. At the same time, wine preferences have become more sophisticated and diverse over the years [27], resulting in an increasing demand of wine *connoisseurs* for premium quality and unique products, priced accordingly [18,25].

Due to their history, pedigree, small scale and traditional forms of production, wines classified as HWP might have a natural advantage in being perceived as singular and authentic products. But at the same time, they remain largely unfamiliar to most consumers, even those typically knowledgeable about Portuguese wines, and are often singular, or downright peculiar products in terms of their sensory profile, packaging and distribution channels [3]. Their increased market acceptance is therefore likely contingent on the adequate provision of contextual knowledge, enabling their proper degustation and the appreciation of their unique features, including their regional character and heritage status. Similar strengths and weaknesses conditioning the successful market repositioning of ancient, high-quality wines have been identified by Chironi and colleagues [25].

Moreover, studies on the product, individual and situational factors affecting how consumers understand and judge the authenticity of food and wine brands remains scarce [14,37], focusing mainly on the perceptions of British, Australian or North American consumers [15,16,23]. Recent studies examining the marketing communication activities of “heroic wines” from Sicily are a notable exception [9,11]. Importantly, research has not yet explored the role played by product knowledge in shaping consumers’ perceptions of wine authenticity and related attributes.

The present study aimed to (1) identify the authenticity attributes consumers associate with wines classified as HWP; (2) assess the relative strength of these associations; and (3) examine how this varied across groups of consumers with different levels of wine knowledge. To this end, we conducted interviews with Portuguese wine producers and consumers, as well as an online questionnaire with a large, convenience sample of the latter. Based on findings, we derived relevant implications for potential marketing strategies promoting a more widespread sale and consumption of wines classified as HWP, thereby also contributing to help preserve part of our country’s cultural food heritage.

## 2. Method

### 2.1. Interview Samples

Complementing an initial review of aforementioned studies on food and wine brand authenticity cues, semi-structured interviews with Portuguese wine producers and adult consumers were first conducted. Their purpose was to identify the range of attributes Portuguese consumers were likely to associate with a wine classified as HWP. Wine producers (*n* = 3) were sought among AVHP members and their contacts. They had different levels of involvement with HWP production (from none to being a current producer, or a producer in the past), but were all familiar with, and knowledgeable about, such wines. In addition, a convenience sample of Portuguese wine consumers (*n* = 12) was assembled among authors’ contacts. Care was taken to interview individuals diverse in sex, age and education, with varied wine drinking habits and knowledge. Given the very low market penetration of wines classified as HWP, we expected most Portuguese consumers to be fairly naïve about them. Accordingly, it was possible to gather that interviewed consumers had little to no prior knowledge about historical wines in general, or specifically about HWP classification, and that they had never heard about AVHP before.

### 2.2. Interview Script

Both wine producers and consumers were first asked to enumerate characteristics they would spontaneously associate with a wine classified as HWP and, subsequently, requested to discuss the relative importance of these attributes to their understanding of, and opinion about, such classification. They were also asked to mention any wines they knew that could, in their opinion, be classified as HWP, and explain what differentiated these from others that did not warrant such classification.

### 2.3. Analysis of Interviews

Two authors transcribed audio recordings of the interviews in full, and independently content-analyzed anonymized transcriptions in order to identify and categorize all excerpts that could constitute authenticity attributes of a wine classified as HWP, according to the dimensions of brand authenticity uncovered by previous research [15,17,18,22]. Divergences in categorization were discussed and solved by consensus, and a final list of HWP classification attributes generated for inclusion in the online questionnaire.

### 2.4. Questionnaire Sample and Administration

A convenience sample of adult Portuguese residents who were wine consumers was surveyed about how strongly they associated these attributes with a wine marketed with an HWP classification. To this end, an anonymous online questionnaire ran for two full months on QuestionPro online survey software (https://www.questionpro.com/us (accessed on 29 April 2021)). A campus paid license was used to ensure the survey ran on mobile devices and resulting data could be readily exported to statistical software for further analysis. A link to the questionnaire was snowballed by e-mail, resulting in 1089 individuals accessing it, 801 filling it in and 641 completing it with valid answers (59% valid response rate).

### 2.5. Questionnaire Design

The online questionnaire first asked respondents how often they consumed wine (1 = Never, 2 = Less than once per month, 3 = At least once per month, 4 = Two to three times per month, 5 = At least once per week, 6 = Two to three times per week, 7 = Daily) to filter out those that never consumed this product. Since wine consumption relies heavily on occasion [41], respondents were also asked where they consumed wine more often (1 = at home, 2 = at the home of relatives or friends, 3 = in restaurants or pubs, 4 = at other places). Next, their wine knowledge was assessed by measuring their degree of awareness of, and exposure to, two Portuguese (*Baga* and *Castelão*) and two foreign (*Pinot Noir* and *Shiraz*) grape varieties. (1 = Never heard of it, 2 = Have heard of it, but cannot recall ever drinking wine made from it, 3 = Have heard of it and can recall drinking wine made from it at least once, 4 = Have heard of it and can recall drinking wine made from it several times). Previous studies of alcoholic beverage knowledge among consumers, particularly table wine knowledge, have shown that questions assessing familiarity with grape varieties used nationally (versus internationally) in winemaking have good ability to discriminate between novice and expert consumers [41,45,46]. *Castelão* and *Shiraz* are furthermore among the most planted native and international wine grape varieties in Portugal, respectively, whereas *Baga* and *Pinot Noir* are less popular. Respondents then rated the extent to which they associated the attributes identified from the literature and interview analysis to a wine marketed with an HWP classification (Likert scale 1 = Not at all associated, 6 = Very much associated). Finally, they indicated the extent to which they expected to find an HWP on sale at different distribution channels (hypermarkets, supermarkets, neighborhood grocery stores, wine stores, online, wine clubs, wineries), and to drink an HWP at different locations (home, home of relatives or friends, restaurants or bars, ceremonies or special events, wine bars or taverns, wine fairs, festivals or similar events, hotels or similar establishments, wine tours or winery visits, events held by national or local authorities) (1 = Never, 6 = Always).

### 2.6. Analysis of Questionnaire Responses

A factor analysis of respondents’ ratings of HWP attributes (Kaiser–Meyer–Olkin measure of sampling adequacy = 920; Bartlett’s test of sphericity x^2^(171) = 6652.38, *p* < 0.001); factor extraction: Principal Axis Factoring with PROMAX rotation) was conducted to identify underlying dimensions of brand authenticity. Mean authenticity dimension scores were computed for each respondent based on his/her ratings of attributes loading primarily in each factor, with high scores indicating strong associations. Factor regression scores subsequently underwent Hierarchical Cluster Analysis, using Euclidean distances and the Ward’s Agglomeration method, to identify and profile groups of respondents with similar patterns of associations. Means of authenticity dimension scores were compared across clusters using Welch’s one-way ANOVA with a Games–Howell post-hoc test.

A factor analysis of the extent to which respondents expected to find an HWP on sale at different distribution channels (Kaiser–Meyer–Olkin measure of sampling adequacy = 0.657; Bartlett’s test of sphericity x^2^(21) = 2049.78, *p* < 0.001); factor extraction: Principal Axis Factoring with PROMAX rotation) was conducted to identify main types of channels. Means of channels scores were computed for each respondent based on ratings of type of channel loading primarily on each factor, with high scores indicating high expectations to find an HWP on sale at the channel. Means of channel scores were compared across clusters using Welch’s one-way ANOVA with a Games–Howell post-hoc test.

A factor analysis of the extent to which respondents expected to drink an HWP at different locations (Kaiser–Meyer–Olkin measure of sampling adequacy = 0.821; Bartlett’s test of sphericity x^2^(36) = 2227.00, *p* < 0.001); factor extraction: Principal Axis Factoring with PROMAX rotation) was conducted to identify underlying main types of locations. Means of location scores were computed for each respondent based on ratings of type of location loading primarily on each factor, with high scores indicating high expectations to drink an HWP at the location. Means of location scores were compared across clusters using Welch’s one-way ANOVA with a Games–Howell post-hoc test.

A Cronbach’s α reliability coefficient was computed from pairwise correlations between ratings of familiarity with *Baga*, *Castelão*, *Pinot Noir* and *Shiraz* grape varieties, to assess their internal consistency as an indicator of respondents’ wine knowledge.

The distributions of respondents’ frequencies and most usual places of wine consumption, sex and education level were compared across clusters using Chi-square tests. Means of the age of respondents were compared across clusters using Fisher’s one-way ANOVA with Bonferroni post-hoc tests.

All statistical analysis were run in SPSS version 25 (IBM, Armonk, NY, USA).

## 3. Results

Analysis of the transcripts of the semi-structured interviews uncovered ten attributes associated with a wine classified as HWP by wine producers and nine by consumers (Table 1).

Questionnaire respondents were mainly male (56%), married (60%) and residing in Lisbon (48%), with a university undergraduate degree (57%), a paid job (78%) and an average age of 41 years. About half of them (48%) consumed wine at least once a week; slightly over half of them (51%) usually consumed wine at home. The median of their familiarity with varieties *Baga* and *Pinot Noir* was 2 = Have heard of it, but cannot recall ever drinking wine made from it, and with varieties *Castelão* and Shiraz was 3 = Have heard of it and can recall drinking wine made from it at least once (Appendix A). Cronbach’s α for familiarity ratings was 0.83, indicating a good internal consistency of corresponding questions as indicator of respondents’ wine knowledge.

Factor analysis of HWP classification attributes yielded four dimensions with Eigen value above 1 which, together, explained 65% of total variance (Table 2). All rotated attributes loaded higher than 0.35 in one factor, but only one loaded higher than 0.35 in two factors. The first dimension, explaining 42% of variance, encompassed five attributes related to ancientness, ageing and heritage, and was thus, accordingly, labelled as “time and tradition”. The second one entailed attributes associated with quality, reputation and accolades, and was hence titled “quality and authentication”; it explained just 9% of variance. The third dimension, explaining 7% of variance, encompassed attributes linked to the *pedigree* of region of origin, unique taste, vintage character and economic benefits, and was therefore labelled as “uniqueness and relationship to place”. The fourth and final one, representing 6% of variance, was composed of attributes related to product and communication strategy, production characteristics and costs, being, accordingly, named “production and marketing”. This dimension entailed attributes that had mostly been mentioned by wine producers during interviews. On average, “uniqueness and relationship to place” was the authenticity dimension most closely associated with HWP, followed by “quality and reputation” and “time and tradition”. “Production and marketing” was the least associated.

Factor analysis of respondents’ expectations to find HWP for sale at different distribution channels yielded two dimensions with Eigen values above 1 which, together, explained 67% of total variance. All rotated channels loaded higher than 0.35% in a single dimension (Appendix A). The first one entailed wine clubs, wine stores, wineries and online stores, and was thus labelled “specialty stores”; the second one comprised supermarkets, hypermarkets and neighborhood grocery stores, and was hence named “general retail”. On average, respondents expected to find HWP for sale more at specialty stores than in general retail.

Factor analysis of respondents’ expectations to drink HWP at different locations yielded three dimensions with Eigen values above 0.95 which, together, explained 70% of total variance. All rotated locations loaded higher than 0.35% in a single dimension (Appendix A). The first dimension comprised wine tours or winery visits, events held by national or local authorities and wine fairs, festivals or similar events, and was thus labelled “wineries and wine events”. The second one consisted of drinking wine at home or at the homes of relatives or friends and was hence named “home”. The third and final one entailed restaurants or bars, hotels or similar establishments, ceremonies or special events, and wine bars or taverns, and was thus titled “HORECA” (Hotels, Restaurants and Catering). On average, respondents expected to drink an HWP more at wineries, wine events or at home than at HORECA establishments.

A satisfactory, three-cluster solution was obtained from brand authenticity dimension regression scores, with the smallest cluster comprising 153 respondents, the middle-sized one 224 and the largest one 264 (Table 3).

The dimensions of authenticity associated with HWP by respondents in the smallest cluster were first “uniqueness and relationship to place”, followed closely by “quality and authentication”. “Time and tradition” and “production and marketing” were both less closely associated with this wine concept. In view of this, they were collectively labelled *Heritage Gatekeepers*. Consumers in this cluster exhibited the highest level of wine knowledge, entailed the highest proportion of respondents consuming wine daily and of those consuming it most commonly at home. They also expected to find HWP for sale more at specialty stores than general retail. Differences in expectations regarding the two types of channels were nevertheless smaller (less than 1 scale point) than those observed for the other two clusters (circa 2 scale points). Moreover, they expected to drink HWP more at wineries, wine events or at home than at HORECA establishments. Again, differences in expectations between the former and the later were smaller than in the cases of the two other clusters. Lastly, *Heritage Gatekeepers* were more often male, generally older and more frequently held a graduate diploma than respondents in the other two clusters.

The dimensions of authenticity associated mostly with HWP by respondents in the middle-sized cluster were “quality and authentication”, followed closely by “uniqueness and relationship to place”. However, they also closely associated “time and tradition” with this concept, contrary to *Heritage Gatekeepers*. The difference of their average associations with “production and marketing” was also much higher (over 1 scale point). In view of this, respondents in this cluster were collectively named *Tradition Revivalists*. They exhibited the lowest level of wine knowledge of all respondents and entailed the highest proportion of those consuming wine less than once per month. But, similarly to *Heritage Gatekeepers*, this cluster entailed a high proportion of respondents drinking wine most commonly at home. *Tradition Revivalists* were the respondents most expecting to find HWP for sale at specialty stores, and the least expecting to find them at general retail. They were also the respondents most expecting to drink these wines at HORECA establishments, comparatively to the other two clusters. Lastly, they were also more often female, held a high school degree the most often, and a university degree the least often.

The dimension of authenticity associated mostly to HWP by respondents in the largest cluster was “uniqueness and relationship to place”. Respondents in this cluster associated “production and marketing” with this wine concept the least. Associations with the remaining two dimensions of authenticity were similarly moderate. Consequently, this cluster of respondents was named *Aspirational Explorers*. They exhibited a level of wine knowledge lower than *Heritages Gatekeepers*, but still significantly higher than that of *Tradition Revivalists*. Nevertheless, they exhibited the lowest proportion of respondents drinking wine daily of all three clusters, as well as the lowest of individuals consuming this product at home. *Aspirational Explorers* expected to find HWP for sale in general retail the least of all three clusters and expected to drink this type of wine at HORECA establishments or at home significantly less than individuals in the remaining clusters. Finally, these respondents were, on average, the same age as *Tradition Revivalists*, but less likely to be female or hold only a high-school diploma.

## 4. Discussion

Results of the semi-structured interviews indicated the existence of large discrepancies between wine producers’ and consumers’ perceptions of a wine classified as HWP. Expectedly, wine producers’ perceptions related mainly to production and marketing features. Earlier studies of the strategies used by luxury wine producers to craft and maintain brand authenticity [16,17,18] also highlight their concerns for the financial performance of brands over the years. A good performance is essential to ensure the status of brands and maintain the ability to employ close-to-traditional production methods, which limit production scale and increase costs. At the same time, luxury wine producers must downplay their business and marketing expertise, as well as their inherent commercial motivations, in order to put forward an image of genuineness and sincerity. The success of such strategies appear to rely, to a large extent, on establishing strong connections to place of origin and, in this way, create yet another point of singularity and distinction to enhance perceptions of authenticity [14,19]. Accordingly, wine producers’ perceptions of the unique or authentic character of wines classified as HWP, elicited during interviews, appeared to be closely linked to evaluations of their region of origin and corresponding *pedigree* in winemaking.

Consumer perceptions’, on the other hand, denoted a more literal understanding of the word “historic”, as in “old” or “ancient” [23] or, in the particular case of wines, “aged”, possibly related to associations established with types of wine with century-old traditions in Portugal (albeit not falling under the classification of HWP), such as Madeira or Port wines. Still, and in line with findings of extant research on food [15,22] and wine authenticity [19], associations reported during interviews also underlined consumers’ reliance on reputational cues or authoritative authentications of the quality and genuineness of a wine classified as HWP. Notably, the accolades of wine critics and foreign market performance seemed to carry more weight than formal classification schemes or labels.

Results from the questionnaire suggest the existence of important effects of product knowledge on consumer perceptions of a wine classified as an HWP. In line with findings from previous research on food and wine authenticity [36,37], both the cultural awareness of Portuguese consumers regarding the variety of wines produced across their country, as well as their direct consumption experience of these products, appear to have affected their perceptions. *Heritage Gatekeepers* rightly exhibited weaker perceptions of a wine classified as HWP than *Tradition Revivalists*, or even *Aspirational Explorers*, since the branding of wines as historic is still quite rare, and yet there are no formal systems or classifications of this kind in place in Portugal or the EU. Nevertheless, they reported the weakest associations of all three clusters in the dimensions “time and tradition” and “production and marketing” and appeared to be the least reliant on external reputation or authentication cues. They associated a wine classified as an HWP mostly to the dimension of “uniqueness and relationship to place”. *Tradition Revivalists*, on the other hand, reported the strongest associations with all four dimensions uncovered, including “production and marketing”. They also represented the segment that appeared to resort mostly to reputational signals and authentication controls to assess product authenticity. Similar differences in how expert versus novice wine consumers process and value wine attribute information from different sources, namely appellations of origin and expert ratings, have been uncovered by experimental economics research [31]. Albeit not quite as wine-savvy as *Heritage Gatekeepers*, *Aspirational Explorers*, as opposed to *Tradition Revivalists*, rightly perceived the HWP classification to be related to products of unique character, conferred more by place of origin than by market or authoritative gatekeeper’s recognition. Likewise, they downplayed the importance of mainstream distribution channels and conventional wine consumption contexts in favor of specialty stores, wineries and wine events in learning about and experiencing the HWP concept. Such findings are well in line with results from knowledge-based segmentation studies of wine consumers [27,36,47]. These point out the important moderating roles played by objective and subjective product knowledge on the use of different product choice cues, sources of information and advice.

## 5. Conclusions and Limitations

This study sought to identify the authenticity attributes consumers associate with wines classified as HWP and assess the relative strength of these associations through the performance of interviews and an online questionnaire with Portuguese wine producers and consumers. Results indicated that, compared to *Aspirational Explorers*, wine *connoisseurs* emerged as a group of *Heritage Gatekeepers* in terms of their perceptions of the HWP brand. Indeed, they associated origin, cultural heritage, quality, reputation and at-home consumption more strongly with wines labelled as HWP, and tradition, wine age, production characteristics and out-of-home consumption less strongly. These results suggest that a high level of wine knowledge is essential for consumers to recognize a wine classified as HWP as a novel, distinct and strictly defined table wine concept. In the absence of that, their claim to *pedigree* and *terroir* will likely be mistaken with that of other wines already marketed in Portugal and abroad for centuries (e.g., vintage Port) but which, by and large, do not share the same traditional production characteristics and thus do not warrant being classified as HWP.

Similar to what was found by previous research [34], *Heritage Gatekeepers* reportedly bought wine more often and appeared to be willing to pay higher prices for it than respondents in the remaining clusters, which exhibited lower wine knowledge. They also seemed to attach more value to the origin of wine, the *terroir* and other production characteristics, such as harvest year, than novices [29,30,31,32,33]. Altogether, the findings presented here reinforce the importance of segmenting potential HWP consumers according to their general level of wine knowledge, and of developing strategies for the positioning of these wines that take the existence of different knowledge-based segments well into account. Ensuing marketing communication activities should thus focus, first and foremost, on teaching wine novices such as *Tradition Revivalists* what the HWP classification stands for, as well as in terms of institutional authenticity [19], and where wines classified as HWP can be purchased. Those targeting savvier wine consumers should focus instead on reinforcing their *terroir*-based authenticity perceptions of the wines falling under this classification.

A limitation of the present study was that it did not ask consumers about the attributes they associated with concrete exemplars of wines classified as HWP, such as those depicted in (Figure 1) and described in (Appendix A). Its main purpose was to identify the kind of positive authenticity cues the HWP classification on its own, without being linked to a specific wine, could evoke on consumers’ minds, so that these could be used by different AVHP producers to better position their wines in the marketplace and communicate appropriately about them to different consumer groups. Future research should focus on the evaluation of different wines labelled as HWP and the characterization of their demand in different countries.

Origin-based, quality certification schemes for food products, such as PDO/PGI, ecological labelling, quality certificates of private organizations or generic denominations of quality, among others, can have positive effects on sales and increase consumers’ willingness to pay for *terroir* products [9,22,31]. In this sense, the HWP classification can be conceived as being an authenticity cue on its own merit, which can function in the market as a heuristic for wine buyers and help improve their perceptions of the quality attributes of the wines falling under it.

Asides from strategic marketing advantages, further development and use of the HWP classification by small, local wine producers could also offer considerable socioeconomic benefits to their regions of origin. Better knowledge about wines classified as HWP, along with a more accurate and favorable appreciation of the attributes that confer them their authentic character and unique quality, should result in an improvement of their sales. It is also important that these wines are adequately differentiated by the market for their merits and do not rely solely on competitive pricing for survival. Given the small size of their domestic and international markets, and the inherent limitations of increasing their scale of production, it is essential that wines classified as HWP are able to maintain their unique character and distinctiveness relatively to other niche or mainstream wine brands, both national and international [25,48]. Last but not least, the social and economic benefits of promoting the HWP classification can have an important spillover effect on the development of the territories in which they are made, on the diffusion of their local culture and the protection of their heritage. As Loureiro [3] states about the broader impact of promoting wines classified as HWP, “it is ever more important to hold on to our wine patrimony and identity, as globalization forces finance its erosion perhaps even without realizing it. It is the duty of Portuguese people devoted to wine and culture to join efforts and defend their wine heritage”.

## Figures and Tables

**Figure 1 foods-10-00979-f001:**
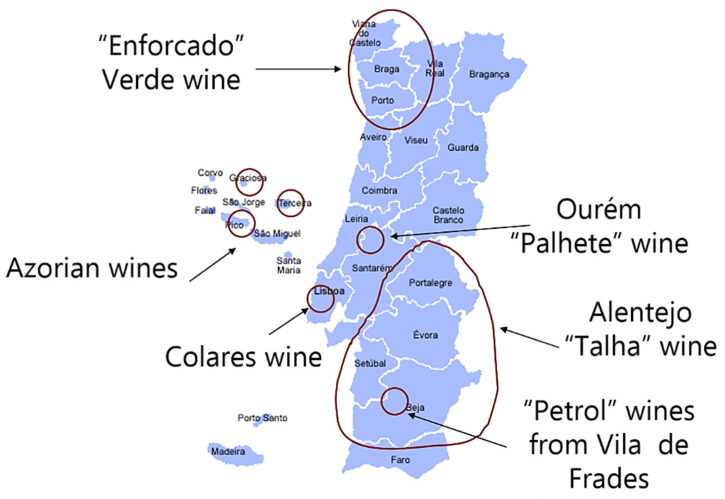
Examples of historical wines and their distribution across Portuguese territory.

**Table 1 foods-10-00979-t001:** Attributes of a wine classified as HWP identified from semi-structured interviews with Portuguese wine producers (*n* = 3) and consumers (*n* = 12).

A Historical Wine of Portugal Is a Wine:	
For Producers	For Consumers
▪ Made only in regions with long tradition in winemaking ▪ Produced and sold for many generations ▪ With a unique taste ▪ With a low alcohol content ▪ Whose quality varies widely across harvest years, depending on climate conditions and grape quality ▪ With a high production cost ▪ That offers good value for money ▪ That contributes to the economic development of its region of origin ▪ Not widely advertised by producers ▪ Made from organically grown grapes	▪ Produced by a very old company ▪ Dating from one or more decades ago ▪ Stored for many years in winemakers’ cellars before being bottled and sold ▪ That should be opened only several years after being bottled ▪ Produced in a special harvest year ▪ Of exceptional quality ▪ Well known and highly reputed ▪ That received prizes and highly favorable reviews from wine critics ▪ Highly appreciated in foreign markets

**Table 2 foods-10-00979-t002:** Factor analysis of the attributes Portuguese wine consumers (*n* = 641) associated to a wine classified as HWP in the online questionnaire.

Dimensions ^1^	Loadings
Time and tradition	
Eigen value = 8.049; Explained variance = 42.4%; Mean ± SD = 3.11 ± 1.30	
Dating from one or more decades ago	0.88
Stored for many years in winemakers’ cellars before being bottled and sold	0.85
Produced by a very old company	0.72
That should be opened only several years after being bottled	0.64
Produced and sold for many generations	0.58
Quality and authentication	
Eigen value = 1.790; Explained variance = 9.4%; Mean ± SD = 3.26 ± 1.22	
Well known and highly reputed	0.82
Highly appreciated in foreign markets	0.77
That received prizes and highly favorable reviews from wine critics	0.76
Of exceptional quality	0.68
Whose quality varies widely across harvest years, depending on climate conditions and grape quality	0.56
Uniqueness and relationship to place	
Eigen value = 1.381; Explained variance = 7.3%; Mean ± SD = 3.49 ± 1.18	
Made only in regions with long tradition in winemaking	0.74
Produced in a special harvest year	0.71
With a unique taste	0.64
That offers good value for money	0.60
That contributes to the economic development of its region of origin	0.57
Production and marketing	
Eigen value = 1.056; Explained variance = 5.6%; Mean ± SD = 2.43 ± 1.02	
Not widely advertised by producers	0.66
Made from organically grown grapes	0.55
With a low alcohol content	0.51
With a high production cost	0.46

^1^ Rated from 1 = “Not at all associated” to 6 = “Very much associated”.

**Table 3 foods-10-00979-t003:** Profiles of groups of questionnaire respondents obtained by clustering HWP authenticity dimension regression scores (*n* = 641).

	Total Sample	Heritage Gatekeepers (24%)	Tradition Revivalists (35%)	Aspirational Explorers (41%)
	Mean ± SD			
HWP authenticity dimensions ^1^				
Time and tradition	3.11 ± 1.30	1.58 ± 0.57 ^A^	4.19 ± 0.93 ^B^	3.07 ± 0.92 ^C^
Quality and authentication	3.26 ± 1.22	1.89 ± 0.75 ^A^	4.40 ± 0.78 ^B^	3.09 ± 0.73 ^C^
Uniqueness and relationship to place	3.49 ± 1.18	2.04 ± 0.73 ^A^	4.36 ± 0.79 ^B^	3.58 ± 0.84 ^C^
Production and marketing	2.43 ± 1.02	1.50 ± 0.59 ^A^	3.20 ± 0.74 ^B^	2.29 ± 0.76 ^C^
Expectations about HWP ^2^				
Likely to be sold at:				
Specialty stores	4.60 ± 1.18	3.85 ± 1.42 ^A^	5.05 ± 0.92 ^B^	4.65 ± 1.01 ^C^
General retail	3.04 ± 1.31	3.15 ± 1.52 ^A^	3.13 ± 1.30 ^A^	2.90 ± 1.18 ^A^
Likely to be drank at:				
Wineries and wine events	3.99 ± 1.26	3.32 ± 1.28 ^A^	4.33 ± 1.19 ^B^	4.09 ± 1.17 ^B^
Home	3.97 ± 1.30	3.28 ± 1.38 ^A^	4.44 ± 1.08 ^B^	3.97 ± 1.24 ^C^
HORECA	3.22 ± 1.04	2.66 ± 1.08 ^A^	3.59 ± 0.98 ^B^	3.23 ± 0.96 ^C^
Wine knowledge ^3^	2.51 ± 1.02	2.71 ± 1.03 ^A^	2.37 ± 0.98 ^B^	2.52 ± 1.03 ^A,B^
Age (years)	40.64 ± 12.21	44.14 ± 11.89 ^A^	39.85 ± 12.14 ^B^	39.25 ± 12.11 ^B^
	Percentage			
Frequency of wine consumption ^†^				
Less than once per month	14.5	10.5 ^a^	19.6 ^b^	12.5 ^a,b^
At least once per month	14.8	13.1 ^a^	15.6 ^a^	15.2 ^a^
Two to three times per month	22.8	21.6 ^a^	19.2 ^a^	26.5 ^a^
At least once per week	18.3	18.3 ^a^	18.3 ^a^	18.2 ^a^
Two to three times per week	12.2	13.1 ^a^	9.4 ^a^	14.0 ^a^
Daily	17.5	23.5 ^a^	17.9 ^a,b^	13.6 ^b^
Most usual place to drink wine ^†^				
Home	51.2	57.5 ^a^	54.5 ^a,b^	44.7 ^b^
Home of relatives or friends	20.7	17.6 ^a^	18.3 ^a^	24.6 ^a^
In restaurants or bars	26.4	23.5 ^a^	25.9 ^a^	28.4 ^a^
Others	1.7	1.3 ^a^	1.3 ^a^	2.3 ^a^
Sex				
Male	55.5	66.9 ^a^	48.6 ^b^	54.8 ^a,b^
Female	44.5	33.1 ^a^	51.4 ^b^	45.2 ^a,b^
Education				
High-school diploma	18.9	12.2 ^a^	26.9 ^b^	16.2 ^a^
Undergraduate diploma	57.1	56.8 ^a^	56.0 ^a^	58.3 ^a^
Graduate diploma	23.9	31.1 ^a^	17.1 ^b^	25.5 ^a,b^

^1^ Rated from 1 = “Not at all associated” to 6 = “Very much associated”. ^2^ Rated from 1 = “Never” to 6 = “Always”. ^3^ Rated 1 = “Never heard of this grape variety”, 2 = “Have heard of this grape variety, but cannot recall ever drinking wine made from it”, 3 = “Have heard of this grape variety and can recall drinking wine made from it at least once”, 4 = “Have heard of this grape variety and can recall drinking wine made from it several times”. Grape varieties: *Baga*, *Castelão*, *Pinot Noir* and *Shiraz*. ^†^ The distribution of frequencies of these variables was not significantly different across clusters. Different superscript capital letters within a row indicate the existence of significant differences (*p* < 0.05) post-hoc in dimension/variable means across clusters. Different superscript lower case letters within a row indicate the existence of significant differences (*p* < 0.05) post-hoc in the proportions of variable classes across clusters.

## Data Availability

Not applicable.

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
