# Peer review of "Historical Wines of Portugal: The Classification, Consumer Associations and Marketing Implications"

_foods, 2021, doi:10.3390/foods10050979_

Round 1

Reviewer 1 Report

Please see the comments inside the PDF. Thank you

Author Response

COMMENTS REVIEWER 1:

 Specific comments

Line 106: please specify the meaning of this acronym.

Thank you for pointing this out, acronyms are fully spelled out in lines 87-88 of the revised manuscript.

Lines 121-123: This statement is not true, there are plenty of studies even in Europe or the united states that analyze how consumers perceive and judge wine and food, both from a sensory point of view and on the basis of credence attributes. As some of the citations are very old (dating back some 20 years), we suggest some reading on this subject that might provide other useful references for the authors to cite in order to provide readers with a complete, and therefore more interesting, overview of the subject (one of these also cites the Porto wine).

Chironi, S., Bacarella, S., Altamore, L., Columba, P., & Ingrassia, M. (2017). Study of product repositioning for the Marsala Vergine DOC wine. International Journal of Entrepreneurship and Small Business, 32(1-2), 118-138.

Altamore, L., Ingrassia, M., Chironi, S., Columba, P., Sortino, G., Vukadin, A., & Bacarella, S. (2018). Pasta experience: Eating with the five senses-A pilot study. AIMS Agriculture and Food, 3(4), 493-520.

Altamore, L., Bacarella, S., Columba, P., Chironi, S., & Ingrassia, M. (2017). The Italian consumers’ preferences for pasta: does environment matter? Chemical Engineering Transactions, 58, 859-864.

Chironi, S., Altamore, L., Columba, P., Bacarella, S., & Ingrassia, M. (2020). Study of Wine Producers’ Marketing Communication in Extreme Territories–Application of the AGIL Scheme to Wineries’ Website Features. Agronomy, 10(5), 721.

Di Vita, G., Chinnici, G., & D'Amico, M. (2014). Clustering attitudes and behaviours of Italian wine consumers. Calitatea, 15(S1), 54.

Pomarici, E., Amato, M., & Vecchio, R. (2016). Environmental friendly wines: a consumer segmentation study. Agriculture and agricultural science procedia, 8, 534-541.

Bruwer, J., & Buller, C. (2012). Consumer behavior insights, consumption dynamics, and segmentation of the Japanese wine market. Journal of International Consumer Marketing, 24(5), 338-355.

We appreciated the suggestion for updating our bibliography as well as the references provided to this end, some of which we cite in the revised manuscript [e.g., in lines 67-68, 229-230, 234-235], to highlight studies on the marketing communication of Sicilian wines. We also revised the text in lines 232-235 to clarify that research on how consumers perceive and evaluate the authenticity of food and wine brands specifically (not the attributes of food and wine in general), remains scarce, and added two recent references supporting this.

 Line 138: Please check the chronological order of the citations, the bibliography must be numbered consecutively without numbering jumps. In addition, references 12 and 13 are missing in the text after number 11.

Thank you for spotting this, the order of citations and numbering of bibliography were fully checked and stand corrected in the revised manuscript.

Line 155: See also

Van Leeuwen, C., & Seguin, G. (2006). The concept of terroir in viticulture. Journal of wine research, 17(1), 1-10.

Chironi, S., Altamore, L., Columba, P., Bacarella, S., & Ingrassia, M. (2020). Study of Wine Producers’ Marketing Communication in Extreme Territories–Application of the AGIL Scheme to Wineries’ Website Features. Agronomy, 10(5), 721.

Ingrassia, M., Altamore, L., Columba, P., Bacarella, S., & Chironi, S. (2018). The communicative power of an extreme territory–the Italian island of Pantelleria and its passito wine. International Journal of Wine Business Research.

Thank you very much for pointing us in the direction of literature on the heroic viticulture in Italy and of an important reference on the concept of terroir, all of which we now discuss in the revised manuscript [e.g., Van Leeuwen & Seguin, 2006 in line 108; Ingrassia et al., 2018 in line 106; Chironi et al. 2020 in lines 234-235].

Line 226: This reference is very dated, please read the comment above about references.

To address this comment, we updated the text and references on the segmentation of consumer wine markets based on product knowledge and involvement in lines 191-205 of the revised manuscript.

Line 249: Capital N in statistics identifies the population or statistical universe of reference, lowercase n the sample size. I think that N=12 is not the population of consumers. Please clarify and explain further.

Apologies for this misunderstanding. A different convention is applied in psychology research, with N denoting sample size and n sub-sample size (e.g., to denote the number of women in a sample); there is no specific notation do denote population size [American Psychological Association Publication Manual (7th ed.), 2020, https://doi.org/10.1037/0000165-000]. This was the convention we used in this paper, which we have now revised throughout to address this comment.

Line 279: Please explain this part of the survey: producers?

Apologies for this lack of clarity. Interviews were conducted with both producers and consumers, but producers were not purposefully sampled to participate in the online questionnaire, only consumers. This section of text just meant to highlight that the attributes evaluated in the questionnaire were those resulting from the literature review and analysis of interview findings earlier performed. We amended the text of line 298 of the revised manuscript to clarify this.

 Line 285: Please add in the methodology a description of the method chosen, in this case Factor Analysis and Cluster Analysis, explaining why it is appropriate for this study.

Thank you for this suggestion. Analysis description and rationale have been added in lines 306-314 of the revised manuscript.

Line 303: Please add a table with Factors with name and % of explained variance and information about eigenvalues.

Thank you for suggesting this. Table 2 of the revised manuscript now depicts the results of the factor analysis.

Line 399: Please add this specification at the beginning of the article (the first time the acronym is mentioned.

Thank you for spotting this.  Acronyms are fully spelled out in lines 87-88 of the revised manuscript; line 460 of the revised manuscript is amended accordingly.

Reviewer 2 Report

Manuscript ID: foods-1138116

Title: Historical Wines of Portugal: the concept, consumer associations and marketing implications

Authors:  Ana Isabel de Almeida Costa, Carla Marano-Marcolini, Manuel Malfeito-Ferreira and Virgílio Loureiro

This paper studied the concept of brand authenticity in wines identifying which attributes consumers and producers associate to historical wines of Portugal and assessing the relative strength. Although the work is interesting and enriches the literature on the subject.

The structure of the paper seems strange to me, I would prefer an introduction that gives a look at the state of the art by removing the part of the description of the wines that I would put into the materials and methods paragraph.

The theoretical background paragraph should be included in the introduction, even leaving the sub-titles, and then at the end broaden the objective of the work already mentioned in lines 120-128.

Abstract should be reduced to 200 words as mentioned in the guidelines for authors.

47 to 120 In my opinion, this part should be put into materials and methods.

133 In my opinion this only applies to the first purchase, then the unknown is reduced even if it remains. Please clarify this point.

247 I would add to this paragraph the part dedicated to the description of historical wines, calling the paragraph Materials and methods. Furthermore, I would insert sub-paragraphs that recall those later used in the Results, perhaps even in the same order. A paragraph dedicated to the statistical analyses is missing. I think the authors have to include a description of the statistical methods used.

249 how were producers and especially consumers chosen? Were they connoisseurs of historical wines? Please clarify this point.

272 Why were these wines chosen? Please, explain that.

263 Please provide a comment on the percentage of valid data from the questionnaire and other information on how the survey was carried out: how long it remained active, with which tool (google modules, other software ...) the questionnaire was implemented and why it was chosen, pros cons…

288 Can you explain a little more? For example, there are attributes mentioned several times, which? ...

Table 1 Why does list  have that numbering? Do you have the possibility to put them in order of importance or according to the number of elicitation of the subjects?

295 Perhaps it would be more interesting to see the distribution of frequencies and indicate the modality with the greatest frequency. I generally disagree with the use of position and variability indices on categorical variables that cannot be approximated to continuous variables. I would say that in materials and methods in a paragraph dedicated to statistical analysis, the statistical analyses made, technical information and the objective should be reported. I would also move all the subsequent technical explanations on factor analysis to this new paragraph in M&M.

302 .35 replace with 0.35

I think it would be important that authors provide a paragraph on the limitations of the work and one with the main conclusions.

Author Response

COMMENTS REVIEWER 2:

General comments

- This paper studied the concept of brand authenticity in wines identifying which attributes consumers and producers associate to historical wines of Portugal and assessing the relative strength. Although the work is interesting and enriches the literature on the subject.

Thank you for your positive appreciation of our work and its contribution to the field.

- The structure of the paper seems strange to me, I would prefer an introduction that gives a look at the state of the art by removing the part of the description of the wines that I would put into the materials and methods paragraph.

Thank you for pointing this out, we reorganized the text in the revised manuscript to adjust its structure towards this comment (please see further details under Specific Comments).

- The theoretical background paragraph should be included in the introduction, even leaving the sub-titles, and then at the end broaden the objective of the work already mentioned in lines 120-128.

Thank you for suggesting this, we merged the theoretical background with the Introduction as you advised in the revised manuscript. We also re-stated the objective of the study in lines 238-245.

- I think it would be important that authors provide a paragraph on the limitations of the work and one with the main conclusions.

We agree, thank you for suggesting this. We added a paragraph on the main conclusions of the work in lines 428-434 of the revised manuscript, and one its main limitations in lines 451-459.

- Abstract should be reduced to 200 words as mentioned in the guidelines for authors.

Thank you for reminding us about this, we shortened the abstract accordingly in the revised manuscript.

 Specific comments

Lines 47-120: In my opinion, this part should be put into materials and methods.

Line 247: I would add to this paragraph the part dedicated to the description of historical wines, calling the paragraph Materials and methods. Furthermore, I would insert sub-paragraphs that recall those later used in the Results, perhaps even in the same order.

We apologize for this misunderstanding. In our interviews and questionnaires, we did not inquire study participants on the attributes associated to any specific HWP described in the Introduction, but rather on those associated to a general, umbrella classification of being “a national historical wine”. Our main goal was to identify the kind of positive authenticity cues this type of classification (on its own, without being linked to a specific wine), would evoke on consumers’ minds, so that these could be later used by different AVHP producers to better position their classified wines in the national market and communicate appropriately about them to different consumer groups.

The description of specific HWP included in the Introduction of the original manuscript was thus only meant to provide (particularly international and non-area readers) with information about, and examples of the type of wines that fall under the classification of HWP, to enhance understanding of its scope. We felt this was necessary given the scarcity and general lack of notoriety of these wines, as well as the still limited number of published works on historic, heroic or similar wine concepts and, to a certain extent, on food and beverage brand authenticity.

Still, we very much appreciated your comments on the appropriate position of this text in the manuscript, as they made us realize that such detailed description could be misleading about what was actually assessed by interviews and questionnaires participants. In view of this, we moved it to a supplementary material file (Table S1) and clarified throughout the revised manuscript text that the object of study were the consumer perceptions of the authenticity attributes of the HWP classification, and of a generic wine classified accordingly, not of specific wines. We also noted the limitations of this approach in the last section of the revised manuscript (lines 451-459).

Line 133: In my opinion this only applies to the first purchase, then the unknown is reduced even if it remains. Please clarify this point.

We agree, thank you for pointing this out. We clarified that uncertainty is indeed higher at product trial, and hence likely to reduce with repeated consumption, in line 217 of the revised manuscript.

Line 247: A paragraph dedicated to the statistical analyses is missing. I think the authors have to include a description of the statistical methods used.

Thank you for pointing out this gap in the methods section. Statistical analysis description and rationale have been added in lines 306-314 of the revised manuscript.

Line 249: How were producers and especially consumers chosen? Were they connoisseurs of historical wines? Please clarify this point.

We appreciate you pointing out that this information was missing in the original manuscript. Wine producers were sought among AVHP members and their contacts. They had different levels of involvement with HWP production (from none to being a current producer, or a producer in the past), but were all familiar with, and knowledgeable about such wines. Wine consumers were sought among authors’ contacts. Care was taken to interview individuals diverse in gender, age and education, with varied wine drinking habits and knowledge. Given the very low market penetration of wines classified as HWP, we expected most Portuguese consumers to be fairly naïve about them. Accordingly, it was possible to gather that interviewed consumers had little to no prior knowledge about historical wines in general, or specifically about the HWP classification, and that they had never heard about AVHP before. This information is now provided in lines 250-260 of the revised manuscript.

Line 263: Please provide a comment on the percentage of valid data from the questionnaire and other information on how the survey was carried out: how long it remained active, with which tool (google modules, other software ...) the questionnaire was implemented and why it was chosen, pros cons…

We appreciate you raising the need to provide further information about survey administration. An online questionnaire ran for two full months on QuestionPro online survey software (https://www.questionpro.com/us). A campus paid license was used to ensure the survey ran on mobile devices and resulting data could be readily exported to statistical software for further analysis. A link to the anonymous questionnaire was snowballed by e-mail, resulting in 1089 individuals accessing it, 801 starting to fill it in and 641 completing it with valid answers (59% valid response rate). This information in now provided in lines 275-278 of the revised manuscript.

Line 272: Why were these wines chosen? Please, explain that.

Previous studies assessing alcoholic beverage knowledge, particularly table wine knowledge, have shown that questions assessing familiarity with grape varieties used nationally (versus internationally) in winemaking have good ability to discriminate between novice and expert consumers (e.g., Schaeffer, 1997; Johnson and Bastian, 2007; Vigar-Ellis et al., 2015). In line with this, we selected two native (Castelão and Baga) and two foreign grape varieties (Pinot Noir and Shiraz) used in Portuguese wine brands that are likely to enjoy very different levels of market awareness. Castelão and Shiraz are among the most planted native and international wine grape varieties in Portugal, respectively, whereas Baga and Pinot Noir are less popular. This explanation has been added in lines 291-297 of the revised manuscript.

Line 288: Can you explain a little more? For example, there are attributes mentioned several times, which? ...

Table 1: Why does list have that numbering? Do you have the possibility to put them in order of importance or according to the number of elicitation of the subjects?

Thank you for raising these points, it made us realize the way we described the main goal of the interviews and how we presented their results in Table 1 could mislead readers. The main purpose of the interviews was to generate the broadest possible range of attributes Portuguese consumers were likely to associate to a wine classified as HWP. We then sought to assess the relative strength of association of each attribute to a wine marketed with the HWP classification by administering all attributes uncovered by interviews to a large sample of consumers in the online questionnaire. Therefore, our content-analysis of interview transcripts was strictly qualitative. With did not count the frequency with which each attribute identified was enumerated by interviewees or whether they attributed more importance to some attributes than others. We assumed that this would emerge, as it did, from the factor analysis of questionnaire results. In the analysis of interviews, we were therefore only concerned with identifying the whole range of attributes that could be associated to a wine classified as HWP. The approach and procedure undertaken to analyze interview results is now described in more detail in lines 266-272 of the revised manuscript.

To further clarify this point, we have also removed the numbering of attributes in Table 1 of the revised manuscript. This was only employed in the original manuscript as a means to abbreviate their future mentions in the text, it did not imply any particular order of importance or frequency of mentions. At the request of another reviewer, we have added a new table (Table 2) to the revised manuscript depicting factor analysis results. From its contents, it is now possible for readers to readily infer the relative strength of association of attributes and dimensions uncovered from questionnaire results.

Line 295: Perhaps it would be more interesting to see the distribution of frequencies and indicate the modality with the greatest frequency. I generally disagree with the use of position and variability indices on categorical variables that cannot be approximated to continuous variables.

We agree, thank you for pointing this out. Indeed, we treated measures of wine knowledge per grape variety as ranked variables in our statistical analysis. Therefore, the mean of means per grape variety is not an adequate or informative descriptive. We have amended this in the revised manuscript by providing the median of wine knowledge per grape variety in the text and adding a table with frequency distributions of wine knowledge per grape variety as a supplementary material file (Table S2).

I would say that in materials and methods in a paragraph dedicated to statistical analysis, the statistical analyses made, technical information and the objective should be reported. I would also move all the subsequent technical explanations on factor analysis to this new paragraph in M&M.

Thank you for highlighting this gap in the methods section. Statistical analysis description and rationale have been now added in lines 306-314 of the revised manuscript, including the factor analysis details earlier presented in the results section.

Line 302: .35 replace with 0.35

Thank you for spotting this typo, it is amended in line 334 of the revised manuscript.
